# Large Uncertainties in $CO_2$ Water–Air Outgassing Estimation with Gas Exchange Coefficient $K_T$ for a Large Lowland River

Anamika Dristi [1] and Y. Jun Xu [1,2,*]

[1] School of Renewable Natural Resources, Louisiana State University, Baton Rouge, LA 70803, USA; adrist1@lsu.edu

[2] Coastal Studies Institute, Louisiana State University, Baton Rouge, LA 70803, USA

* Correspondence: yjxu@lsu.edu

**Abstract:** Aquatic $CO_2$ emission is typically estimated (i.e., not measured) through a gas exchange balance. Several factors can affect the estimation, primarily flow velocity and wind speed, which can influence a key parameter, the gas exchange coefficient $K_T$ in the balancing approach. However, our knowledge of the uncertainty of predictions using these factors is rather limited. In this study, we conducted a numeric assessment on the impact of river flow velocity and wind speed on $K_T$ and the consequent $CO_2$ emission rate. As a case study, we utilized 3-year (2019–2021) measurements on the partial pressure of dissolved carbon dioxide ($pCO_2$) in one of the world's largest alluvial rivers, the lower Mississippi River, to determine the difference in $CO_2$ emission rate estimated through three approaches: velocity-based $K_T$, wind-based $K_T$, and a constant $K_T$ (i.e., $K_T$ = 4.3 m/day) that has been used for large rivers. Over the 3-year study period, river flow velocity varied from 0.75 ms$^{-1}$ to 1.8 ms$^{-1}$, and wind speed above the water surface fluctuated from 0 ms$^{-1}$ to nearly 5 ms$^{-1}$. Correspondingly, we obtained a velocity-based $K_T$ value of 7.80–22.11 m/day and a wind-speed-based $K_T$ of 0.77–8.40 m/day. Because of the wide variation in $K_T$ values, the estimation of $CO_2$ emission using different approaches resulted in a substantially large difference. The velocity-based $K_T$ method yielded an average $CO_2$ emission rate (FCO$_2$) of 44.36 mmol m$^{-2}$ h$^{-1}$ for the lower Mississippi River over the 3-year study period, varying from 6.8 to 280 mmol m$^{-2}$ h$^{-1}$. In contrast, the wind-based $K_T$ method rendered an average FCO$_2$ of 10.05 mmol m$^{-2}$ h$^{-1}$ with a small range of fluctuation (1.32–53.40 mmol m$^{-2}$ h$^{-1}$,), and the commonly used constant $K_T$ method produced an average FCO$_2$ of 11.64 mmol m$^{-2}$ h$^{-1}$, also in a small range of fluctuation (2.42–56.87 mmol m$^{-2}$ h$^{-1}$). Based on the findings, we conclude that the effect of river channel geometry and flow velocity on $CO_2$ outgassing is still largely underestimated, and the current estimation of global river $CO_2$ emission may bear large uncertainty due to limited spatial coverage of flow conditions and the associated gas exchange variation.

**Keywords:** $CO_2$ emission; gas exchange coefficient; $pCO_2$; riverine carbon; Mississippi River

## 1. Introduction

Rivers transport a large quantity of terrestrial carbon in a variety of forms, including dissolved, particulate, organic, inorganic, and detritus, through dense channels to the world's oceans [1]. In addition to the lateral export of carbon, river water also emits $CO_2$ gas into the atmosphere. Studies have found that many rivers in the world function as a source of $CO_2$ to the atmosphere [2–4], resulting in a large global river $CO_2$ flux to the atmosphere between 230 and 1800 Tg each year [5,6]. The outgassing quantity of carbon likely exceeds the lateral carbon export. Many factors can contribute to this vertical carbon flux, but our knowledge about them is still incomplete. Studies in recent decades have found that several factors can control riverine $CO_2$ emissions, and these factors may be categorized into environmental, biogeochemical, and anthropogenic factors [7]. Biological factors such as organic matter input affect the amount of organic matter entering

rivers from terrestrial ecosystems and influence the rate of microbial decomposition and $CO_2$ production [8]. In-stream processing, such as aquatic respiration, photosynthesis, and mineralization, can determine the amount of $CO_2$ produced and emitted by rivers. However, estimation of the $CO_2$ production is challenging and mostly indirect.

The above natural processes are further influenced by anthropogenic factors, including land use and land cover, which can modify the quantity and quality of organic matter entering rivers, which in turn can influence $CO_2$ emissions [9,10]. Human-made dams, reservoirs, and other water management practices can alter river flow, water temperature, and organic matter availability, impacting stream partial pressure of dissolved $CO_2$ ($pCO_2$) level and emissions [11,12]. Studies have found a strong relationship between the $CO_2$ efflux and the proportion of urban land in the catchment area [13,14]. Environmental factors such as flow velocity, channel morphology, and wind can influence the exchange of $CO_2$ between water and air through turbulence and mixing. These factors are critical in air–water $CO_2$ flux calculation, strongly affecting river carbon budgeting. However, our knowledge of their combined effect is rather limited, especially for large river systems, which contribute a significant volume of $CO_2$ to global net carbon emission [15,16].

The Mississippi–Atchafalaya River system is the largest river system in North America, containing over 41% drainage area of the continental United States and discharging approximately 680 km$^3$ (2% of global annual discharged freshwater to oceans) of water annually into the Gulf of Mexico [17]. Large rivers, such as the Mississippi, collect heterogenic forms of carbon washed away from lands, agricultural fields, residences, treatment plants, and chemical companies through millions of narrow channels that are regulated by flow, rain, temperature, groundwater seepage, and other factors. The more urbanization, industrialization, and agricultural fields, the more carbon fluxes into rivers. Runoffs containing nutrients from cropland, urban land, and treatment plants can enrich nutrients in adjacent rivers, thus speeding up the production of $CO_2$ and carbon loss in river ecosystems [9,18]. According to several studies, the annual DOC export from the lower Mississippi River is 1.5~4.1 Tg [15,19,20], while the estimated annual DIC export is 12.25~13.6 Tg [15,21], which is thought to be grown by 40% over the last 100 years due to land management practices [22]. As a result of land use, seasonal flow differences, and various physical properties of the river, the partial pressure of $CO_2$ ($pCO_2$) in rivers fluctuates spatially and seasonally [4]. Therefore, the outgassing $CO_2$ into the air from the Mississippi River varies largely.

Despite the fact that rivers contribute a significant quantity of $CO_2$ outgassing to net $CO_2$ release in the atmosphere, the amount varies substantially depending on a variety of factors, including environmental (e.g., wind, flow velocity) and anthropogenic influences (e.g., land use, excess nutrient input). The most used method for estimating $CO_2$ outgassing from rivers is a gas exchange balance equation (Equation (1)) [23]:

$$FCO_2 = K_T K_H (pCO_{2\ water} - pCO_{2\ air}) \tag{1}$$

where $K_T$ is the gas exchange coefficient positively correlated with the temperature normalized (at 20 °C) $K_{600}$ values; the flux depends on the $K_T$ values. $pCO_{2water}$ is the partial pressure of dissolved carbon dioxide in water. The partial pressure of carbon dioxide in the air, $pCO_{2air}$, is also utilized in Equation (1) and is set at 410 μatm in this study (Tans et al. 2021) [24]. $K_H$ is the solubility constant measured in mol L$^{-1}$ atm$^{-1}$. The calculation of $K_H$ was performed using Equation (2) [25]:

$$lnK_H = A_1 + A_2 \left( \frac{100}{T} \right) + A_3\, ln \left(\frac{T}{100}\right) + S \left[ B_1 + B_2 \left(\frac{T}{100}\right) + B_3 \left(\frac{T}{100}\right)^2 \right] \tag{2}$$

where $A_1$, $A_2$, $A_3$, $B_1$, $B_2$, and $B_3$ are $-58.0931$, $90.5069$, $22.2940$, $0.027766$, $0.025888$, and $0.0050578$, respectively; $T$ and $S$ represent the absolute temperature of water in Kelvin and the salinity in parts per thousand, which was set to 0 (i.e., $S = 0$) because the lower Mississippi River at Baton Rouge is considered completely freshwater.

$K_T$ is the primary driver of $CO_2$ emissions from rivers. Even though the $CO_2$ content in the river is lower, $CO_2$ emissions can still be substantial due to the higher value of the gas exchange coefficient $K_T$ [26,27]. As a result, variations in the $K_T$ selection can result in major shifts in the estimated $CO_2$ flux from the river. For instance, the study by Xu and Xu utilized a $K_T$ value of 4.3 m/day, resulting in $CO_2$ fluxes of 777 g C m$^{-2}$ yr$^{-1}$ [28], while Potter and Xu [29], Reiman and Xu [15], and Dubois et al. [20] used 3.9 m/day, resulting in carbon fluxes of 864 g C m$^{-2}$ yr$^{-1}$, 654 g C m$^{-2}$ yr$^{-1}$, and 1036 g C m$^{-2}$ yr$^{-1}$, respectively. This wide range of carbon fluxes emphasizes the importance of selecting an appropriate gas transfer coefficient, $K_T$. There are generally three methods for calculating $K_T$: (1) based on flow velocity, (2) based on wind speed, and (3) using a constant $K_T$. $CO_2$ outgassing studies from the Amazon River [1,30] and the Mississippi River [4] showed a significant positive linear relationship between $p CO_2$ and river discharge in tidal rivers. In contrast, this relationship is negative in nontidal rivers, with a positive $p CO_2$ and wind relationship. It is relatively well-established knowledge that $k_{600}$ is governed by a multitude of physical factors, particularly river flow velocity, wind speed, stream slope, and water depth in open waters such as large rivers and estuaries [7,31,32]. Li et al. calculated $CO_2$ flux using both $K_T$ values derived from the floating chamber method and the water velocity-dependent model from the river data where a large flux difference is found [33]. Alin et al. found that $k_{600}$ and water current velocity were positively and significantly correlated in small rivers and streams and expected a positive relationship between $k_{600}$ and water current velocity in large rivers if water current velocity data had been collected at the same time and place as the $k$ measurements [7]. However, it is not clear how differently the flux can vary due to the velocity-based $K_T$ in rivers.

Wind speed has also been found to be one of the primary forces of the aqueous boundary layer that controls gas exchange and greatly affects $K_T$. Wanninkhof and McGillis demonstrated a long-term cubic relationship between air–sea gas exchange and wind speed [34], while other studies later found a strong linear relationship between gas transfer velocity and wind speed in large rivers [7]. Additionally, another $K_T$ value of 4.3 m/day is also being used, which has earlier been found to be typical for large tropical lowland rivers [7] and has also been applied in other studies to consider the corresponding result of a conservative outgassing estimate [28]. Until now, to our best knowledge, no study exists that compares $CO_2$ outgassing with velocity-based, wind-speed-based, and constant $K_T$ for large rivers. It is also not clear whether other environmental factors would affect the estimation of $K_T$, such as discharge and temperature. $K_T$ value can strongly affect $CO_2$ emission, but field measurements of $K_T$ for large rivers are both technically difficult and constrained by funding and human resources.

With the above introduction in mind, this study was conducted to analyze three common approaches in determining $K_T$ and their impact on $CO_2$ outgassing estimation. As a case study, we utilized 3-year field measurements on the partial pressure of dissolved carbon dioxide ($p CO_2$) in the lower Mississippi River and other relevant parameters. We aimed to test the hypothesis that the common $K_T$ determination approaches large uncertainties in FCO$_2$ estimation. Specifically, this study aimed to (1) determine the variation in $K_T$ based on velocity and wind speed, (2) investigate the impact of the variation on FCO$_2$ estimation, (3) assess the difference in FCO$_2$ estimation of the two methods with setting $K_T$ as a constant, and (4) analyze the possible correlation of $K_T$ with other ambient parameters. The goal of this study was to deliver up-to-date information on estimating gas transfer coefficient $K_T$ based on river characteristics and weather conditions, allowing for more accurate $CO_2$ outgassing estimation from heterotrophic rivers.

## 2. Methodology

### 2.1. Study Site

This research was carried out for the lower Mississippi River at Baton Rouge (30°26′44.4″ N, 91°11′29.6″ W), LA, USA (Figure 1). The Mississippi River drains an area of 3.2 million km$^2$, equivalent to approximately 41% of the contiguous United States. Over

the past four decades, the river discharged an average of 673 km$^3$ of freshwater annually into the Gulf of Mexico via its mainstream channel and 474 km$^3$ [35] via its distributary, the Atchafalaya River (199 km$^3$) [36]. The site's proximity to a USGS gauging station, short distance to the Gulf of Mexico (approximately 368 km), and its expansive drainage capturing water from nearly the entire Mississippi River Basin (2.92 million km$^2$) make it an ideal location for analyzing carbon export and studying carbon dynamics in the region.

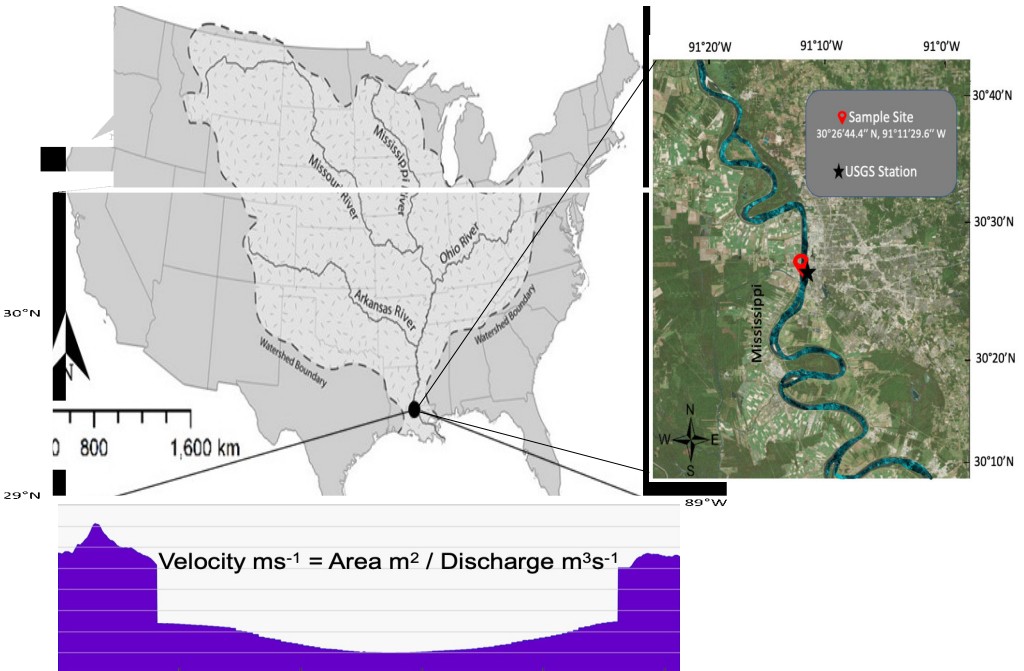

**Figure 1.** The Mississippi River Basin in the United States, the study location at Baton Rouge in Louisiana, and the river channel geometry. The Mississippi River at the location is leveed on both sides. Average flow velocity was computed based on the formula using discharge and the channel cross-sectional area.

The city of Baton Rouge lies 16–18 m above sea level. The elevation of the lower Mississippi River channel at Baton Rouge is approximately 2.15 m above sea level. Long-term annual temperature in the Baton Rouge area was reported to be 20 °C, with monthly averages ranging from 11 °C in the coldest month (January) to 28 °C in the warmest month (July) [37]. Long-term annual precipitation in the area was reported to be about 1477 mm, ranging from 159 mm in July to 81 mm in October. The prevailing conditions in this region can be characterized by a humid and subtropical climate.

### 2.2. Data Collection

This study utilized a series of data from field measurements, including the partial pressure of dissolved carbon dioxide ($p$CO$_2$), water temperature (T), wind speed, daily average discharge, and stage records at the US Geological Survey (USGS) Baton Rouge gauge station (station# 07374000). Part of this data was published by (Xu and Xu) and (Potter and Xu) [28,29]. The measurements of $p$CO$_2$ were used to compute CO$_2$ outgassing from the river surface. More information on field collection can be found in the two publications. Briefly, field measurements and sampling were conducted on a local ferry about 80 m into the Mississippi River monthly from January 2019 to December 2021. All measurements were performed between 9:00 and 9:30 a.m. US Central Standard Time (CST) in order to keep possible effects from variations in solar radiation and river respiration. During each sampling trip, partial pressure of dissolved carbon dioxide ($p$CO$_2$) was measured with a C-SenseTM sensor (Turner Designs, San Jose, CA, USA); other water parameters and DO were recorded with a YSI 556 multi-probe meter (YSI Inc.,

Yellow, Springs, OH, USA). Wind speeds at the sampling time were taken using a Kestrel 5500 Weather Meter (Nielsen-Kellerman Company, Boothwayn, PA, USA). Daily average discharge and water temperature records were obtained from the USGS.

The discharge records were used to determine the average river flow velocity for estimating the gas exchange coefficient. Therefore, the river geometry information was needed. The usual width of the Mississippi River in Baton Rouge is approximately 1200 m [38], but the sampling site's width is approximately 600 m [28]. The cross-section area and concurrent river discharge in the Mississippi River were used to compute the velocity for each sampling day. Equations, which were adopted from the stage cross-section curve in the lower Mississippi River, were used to calculate the cross-section area and velocity (Equations (3) and (4)):

$$Cross\ Section\ A\ (\text{m}^2) = 6836.758242\ \text{m}^2 + (938.441\ \text{m} \times River\ Stage\ \text{m}) \tag{3}$$

$$Velocity\ (\text{ms}^{-1}) = \frac{Cross\ Section\ A\ (\text{m}^2)}{Daily\ Discharge\ (\text{m}^3\text{s}^{-1})} \tag{4}$$

Part of the data used in this study, which was collected from January 2019 to December 2021, was published by Xu and Xu to quantify lateral and vertical carbon transport of the Mississippi River during its 2019 mega-flood [28].

### 2.3. Estimations of Gas Exchange Coefficient $K_T$

For $CO_2$ outgassing flux calculation, this study took three different approaches to determine $K_T$: (1) wind speed, (2) velocity, and (3) constant. Wind speed and penetrative convection are the dominant controls on surface turbulence and, thus, gas transfer in lakes and the open ocean [39], while $K_{600}$ has traditionally been modeled in stream environments as a function of stream depth, water current velocity, discharge, and slope. Alin et al. demonstrated different calculation approaches for the critical factor of $K_T$, $K_{600}$, where wind speed ($\mu_{10}$ in $\text{ms}^{-1}$) was used as a driving force given in Equation (5) [7].

$$K_{600} = 4.46 + 7.11\ \mu_{10} \tag{5}$$

In this study, wind speed was measured at a height of approximately 8–10 m above the river water surface.

Stream flow velocity (w in $\text{ms}^{-1}$) was also used as a driving force in many river studies, as given in Equation (6) [7]:

$$K_{600} = 13.82 + 0.35\ \text{w} \tag{6}$$

In this study, for each sampling date, the velocity was calculated using the cross-section area of the river as well as the concurrent river discharge in the Mississippi River at Baton Rouge (Equation (4)). In order to determine the area of the cross-section, Equation (3) was utilized, which was taken from the stage cross-section curve calculated in the lower Mississippi River. $K_{600}$ is the normalized $K$ value at 20° Celsius (Equation (7)) [7], and finally, $K_T$ was computed with Equation (8):

$$K_{600} = K_T \left(\frac{600}{S_{CT}}\right)^{-\frac{1}{2}} \tag{7}$$

$$K_T = K_{600} \left(\frac{S_{CT}}{600}\right)^{-\frac{1}{2}} \tag{8}$$

We calculated three different $K_T$ values from Equation (8) where constant $K_{600}$, velocity-based $K_{600}$, and wind-based $K_{600}$ were used and named the resulting $K_T$ as fixed $K_T$,

velocity-dependent $K_T$, and wind-dependent $K_T$. The Schmidt value for freshwater was calculated as a function of temperature (Equation (9)), T, in degrees Celsius [28].

$$S_{CT} = 1911.1 - 118.11T + 3.4527T^2 - 0.04132T^3 \tag{9}$$

A fixed $K_T$ value of 4.3 m/day was employed as the constant $K_T$ in our study. The selection of a constant $K_T$ was based on prior studies that were used to estimate a consistent $CO_2$ outgassing applicable in similar river systems. According to Alin et al., the $K_T$ value of 4.3 m/day is considered ideal for large tropical lowland rivers [7]. Additionally, the $CO_2$ outgassing from the specific site has been studied in two earlier studies where both used the $K_T$ value of 4.3 m/day [15,28]. In addition, a separate study estimated a gas transfer velocity of 3–4 m/day for large rivers like the Mississippi [6]. As a result, we chose to utilize 4.3 m/day as the constant $K_T$ value in our study.

### 2.4. Statistical Analysis

Statistical analyses used in this study included analysis of variance (ANOVA) and correlation and regression analysis. These analyses were performed to determine the significance of $K_T$ estimated through the flow velocity-based and wind-speed-based methods and the constant value. The statistical test was performed for both $K_T$ and the resulting $FCO_2$ estimates to assess the key variables that drove $CO_2$ outgassing. A multiple-comparison by Tukey HSD test (conf. level = 0.95) was performed pairwise between different $CO_2$ fluxes for different types of $K_T$ values to find out which approach of $K_T$ drove the $CO_2$ flux from the river.

## 3. Results

### 3.1. River Conditions

A large fluctuation in the river conditions was observed in the Mississippi River during the study period from 2019 to 2021 (Figure 2). The river flow ranged from 6938 to 36,812 cubic meters per second ($m^3\,s^{-1}$), with a mean of 20,191 $m^3\,s^{-1}$ ($\pm$9590 $m^3\,s^{-1}$). The river flow was high during the winter and spring months, falling to the lowest level in the late summer and fall. These discharge records were grouped into five phases based on the NOAA's river stage classification for the Mississippi River at Baton Rouge: Low, Action, Intermediate, High, and Peak discharge (Table 1). This grouping was deemed to estimate $K_T$ under different flow conditions.

**Table 1.** River discharge ranges for five flow conditions. The Low and Action flow stages were adopted from Joshi and Xu [35], while those for the Intermediate, High, and Peak flow stages were adopted from Rosen and Xu [36].

| | Low | Action | Intermediate | High | Peak |
|---|---|---|---|---|---|
| Flow Stage (m) | <9.8 | 9.8–12.1 | 12.1–14.6 | 14.6–16.8 | >16.8 |
| Discharge Range ($m^3\,s^{-1}$) | <13,000 | 13,000–18,000 | 18,000–25,000 | 25,000–32,000 | >32,000 |

We computed the average daily flow velocity for the three study years (Figure 3) by using the river discharge records and the cross-sectional area (Equation (4)). Daily average velocities for the sampling dates ranged from 0.8 to 1.9 m per second ($ms^{-1}$), with an average of 1.39 $ms^{-1}$ ($\pm$0.44 $ms^{-1}$). Generally, the average daily velocity peaked in winter and spring and declined in late summer and fall, along with the river discharge trend. The geometry of the river channel influenced flow velocities directly by determining the cross-sectional area.

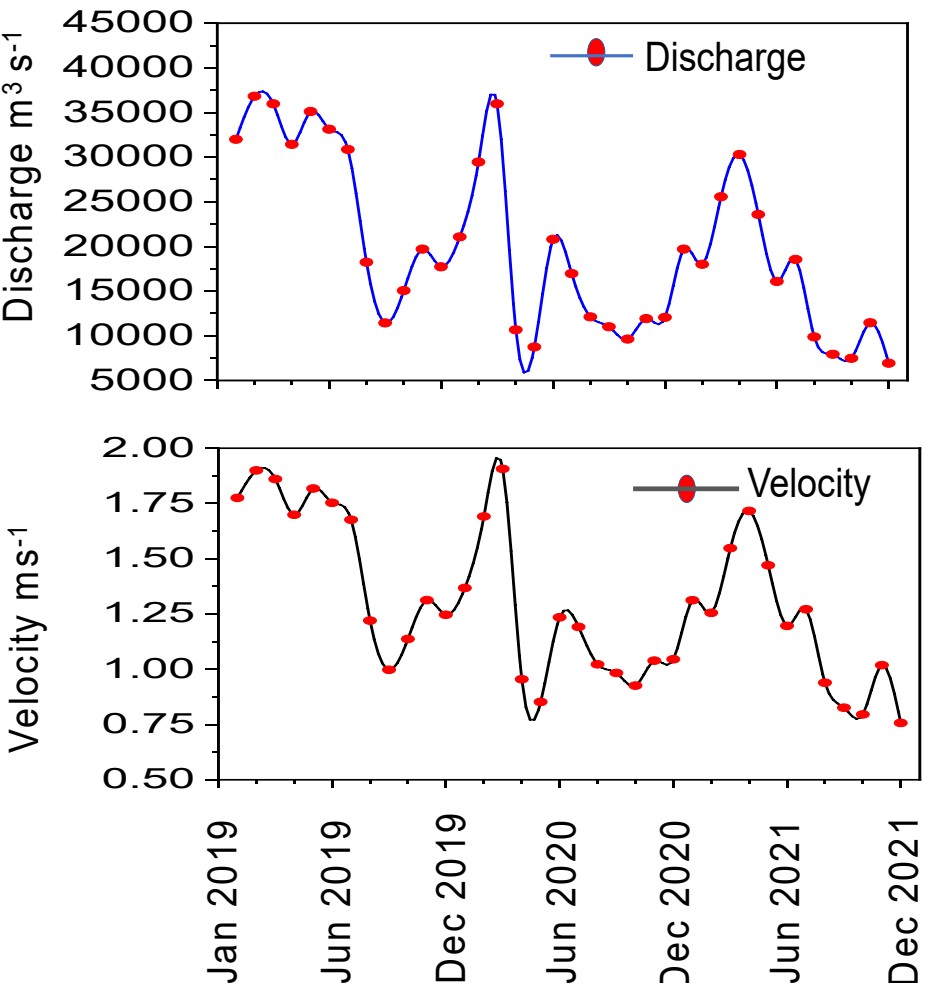

**Figure 2.** Fluctuations of the daily mean river discharge (20,191 $\pm$ 9590 m$^3$ s$^{-1}$; mean $\pm$ SD) and calculated average flow velocity (1.39 $\pm$ 0.33 ms$^{-1}$; mean $\pm$ SD) in the Mississippi River at Baton Rouge in Louisiana, the United States, from January 2019 to December 2021. Solid dots indicate sampling dates.

Over the 3-year study period, the river was exposed to low wind speeds at the sampling dates, varying, however largely, from 0.4 to 4 ms$^{-1}$ with an average of 1.6 ms$^{-1}$ $\pm$ 0.9; mean $\pm$ SD (Figure 3). The variation is ten-fold, i.e., larger than that of the river discharge. The wind speed data were used in Equations (5) and (8) for calculating wind-based $K_T$. During the same period, water temperature in the Mississippi River at Baton Rouge varied widely, ranging from 2.96 °C in January to 29.90 °C in August, with an average of 18.68 °C $\pm$ 7.91; mean $\pm$ SD. The fluctuation of temperature has an effect on the calculation of the Schmidt value (Equation (9)), which was used to calculate $K_T$. The partial pressure of the $CO_2$ and $pCO_2$ levels in the Mississippi River was clearly higher than in the atmosphere, with 90% of $pCO_2$ levels in the river exceeding twice the $CO_2$ concentration in the atmosphere (410 ppm) from January 2019 to December 2021. During the study period, the Mississippi River experienced a wide range of $pCO_2$ concentrations, ranging from 700 to 4350 µatm, with an average of 1885 µatm $\pm$ 830; mean $\pm$ SD (Figure 3). The direction and magnitude of the $CO_2$ discharge are determined by the partial pressure gradient between the dissolved $CO_2$ in the water and the $CO_2$ in the atmosphere. Higher $pCO_2$ in the water has immense outgassing potential. The daily average $pCO_2$ concentrations were utilized to compute the river's $CO_2$ flux.

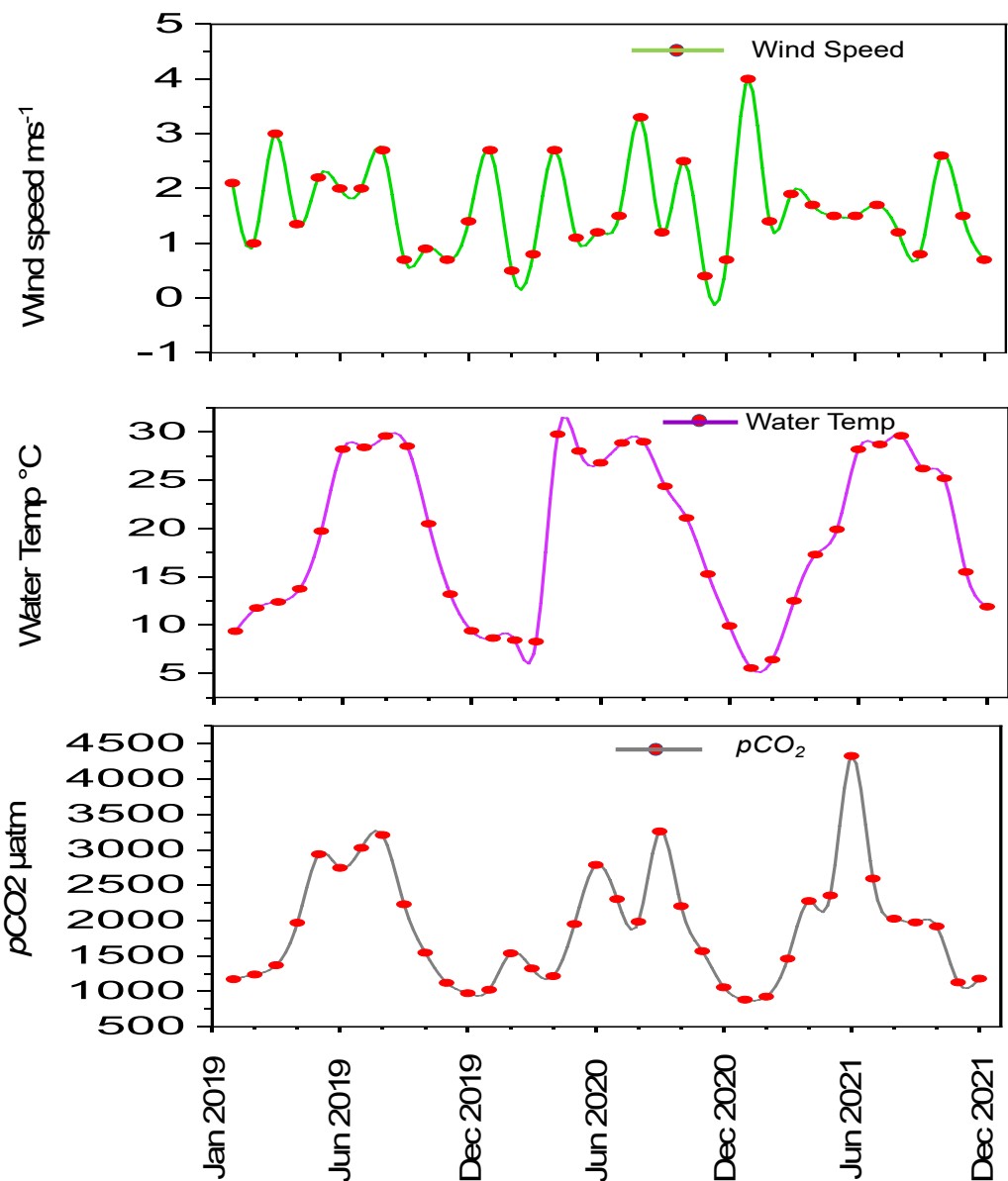

**Figure 3.** Variation of river water temperature (~19 °C ± 8; mean ± SD), wind speed (1.6 ms$^{-1}$ ± 0.9; mean ± SD), and $p$CO$_2$ (1885 µatm ± 830; mean ± SD) in the Mississippi River at Baton Rouge in Louisiana, the United States, from January 2019 to December 2021. Solid dots indicate the sampling date.

### 3.2. Differences in Estimated $K_T$ Values

Using Equations (5) and (8), we computed the gas exchange coefficient based on wind speed. Using Equations (6) and (8), we calculated a velocity-based $K_T$. The wind-based $K_T$ ranged from 0.77 to 8.40 m/day, while the velocity-based $K_T$ ranged from 7.80 to 22 m/day (Table 2). Though both wind- and velocity-based $K_T$ values showed substantial variation during the 3-year study period (Figure 4), this study found average $K_T$ values around 14.61 m/day for the velocity-based method, which is three-fold higher than the other two resulting average $K_T$ values, 4.3 m/day (fixed method) and 3.65 m/day (wind-based method) (Table 2).

**Table 2.** Comparison of $k_{600}$ (m/day) and $K_T$ (m/day) values based on their calculations using flow velocity and wind speed, along with a constant $K_T$.

|  |  | **Constant** | **Velocity-Based** | **Wind-Based** |
|---|---|---|---|---|
| $K_{600}$ | Min |  | 9.67 | 1.07 |
| (m/day) | Max |  | 19.33 | 7.89 |
|  | Mean ± SD |  | 14.34 ± 3.70 | 3.81 ± 1.50 |
| $K_T$ | Min | 4.3 | 7.80 | 0.77 |
| (m/day) | Max | 4.3 | 22.11 | 8.40 |
|  | Mean ± SD | 4.3 | 14.61 ± 3.76 | 3.65 ± 1.60 |

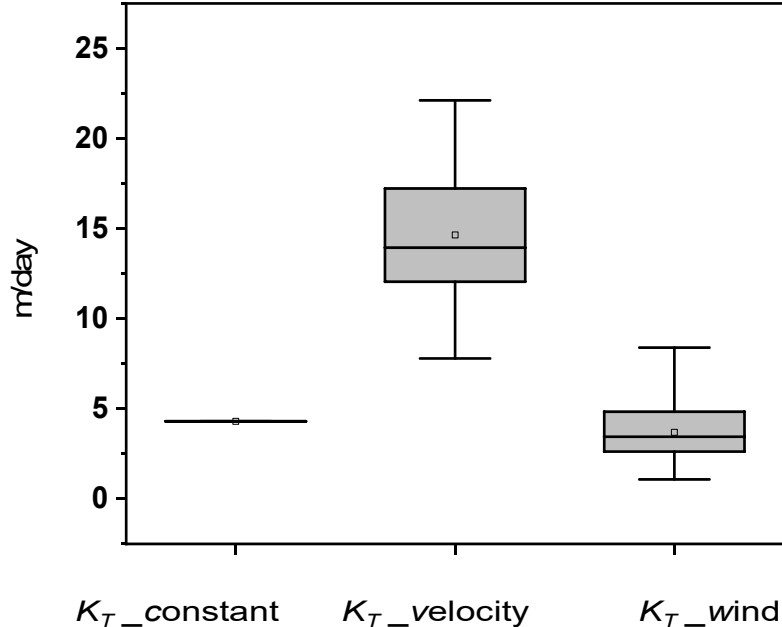

**Figure 4.** Large variation in velocity-based $K_T$ when compared to those with constant $K_T$ and wind-based $K_T$ during the period of 2019–2021. The points inside boxplots indicate the mean values for the parameters.

$K_{600}$ and $K_T$ varied largely among the different estimation methods (Table 2). The $K_{600}$ values obtained from the velocity and wind speed readings were used to calculate the velocity and wind-based $K_T$ (Equations (5), (6), and (8)). The average wind-based $K_T$ (3.65 m/day) was even lower than the constant $K_T$ (4.3 m/day) (Table 2), which is considered a gas transfer coefficient representative of large lowland rivers.

### 3.3. CO₂ Outgassing (FCO₂) Estimation with Velocity-Based and Wind-Based $K_T$

From January 2019 to December 2021, we measured $CO_2$ outgassing (FCO₂) from the Mississippi River using constant, velocity-based, and wind-based $K_T$ methods. As $K_T$ is the driving factor of the outgassing process, the resulting $CO_2$ fluxes exhibit similar trends due to the variations in $K_T$. A wide variation was observed between the $CO_2$ fluxes calculated using the three $K_T$ estimation methods, reflecting the substantial $K_T$ fluctuations.

$CO_2$ fluxes based on the constant $K_T$ approach ranged from 2.42 to 56.87 mmol m$^{-2}$ h$^{-1}$, with an average value of 11.64 ± 8.15 mmol m$^{-2}$ h$^{-1}$ (Table 3). Similarly, the wind-based $K_T$ method yielded $CO_2$ fluxes ranging between 1.32 and 53.39 mmol m$^{-2}$ h$^{-1}$, with an average value of 10.05 ± 8.65 mmol m$^{-2}$ h$^{-1}$ (Table 3). The velocity-based $K_T$ approach exhibited the highest variation in $CO_2$ outgassing, though the constant $K_T$ and wind-based $K_T$ methods had similar variations. During the study period, $CO_2$ fluxes estimated from the velocity-based $K_T$ ranged from 6.80 to 280 mmol m$^{-2}$ h$^{-1}$ (Table 3). We found the average carbon flux calculated from velocity-based $K_T$ (44.36 mmol m$^{-2}$ h$^{-1}$) to be nearly

four times higher than that measured by the constant $K_T$ method (11.60 mmol m$^{-2}$ h$^{-1}$) and the wind-based method (10 mmol m$^{-2}$ h$^{-1}$) (Figure 5). As a result, the choice of the $K_T$ estimation method significantly impacts the magnitude of $CO_2$ outgassing, emphasizing the need for careful selection of $K_T$ for accurate estimation.

**Table 3.** Comparison between three different $CO_2$ outgassing (FCO$_2$) estimated from the constant, velocity-based, and wind-based $K_T$. River discharge ranges for five flow conditions.

|  |  | **Constant** | **Velocity-Based** | **Wind-Based** |
|---|---|---|---|---|
| FCO$_2$ | Min | 2.42 | 6.8 | 1.32 |
| (mmol m$^{-2}$ h$^{-1}$) | Max | 56.87 | 280 | 53.40 |
|  | Mean $\pm$ SD | 11.64 $\pm$ 8.15 | 44.36 $\pm$ 43 | 10.05 $\pm$ 8.65 |

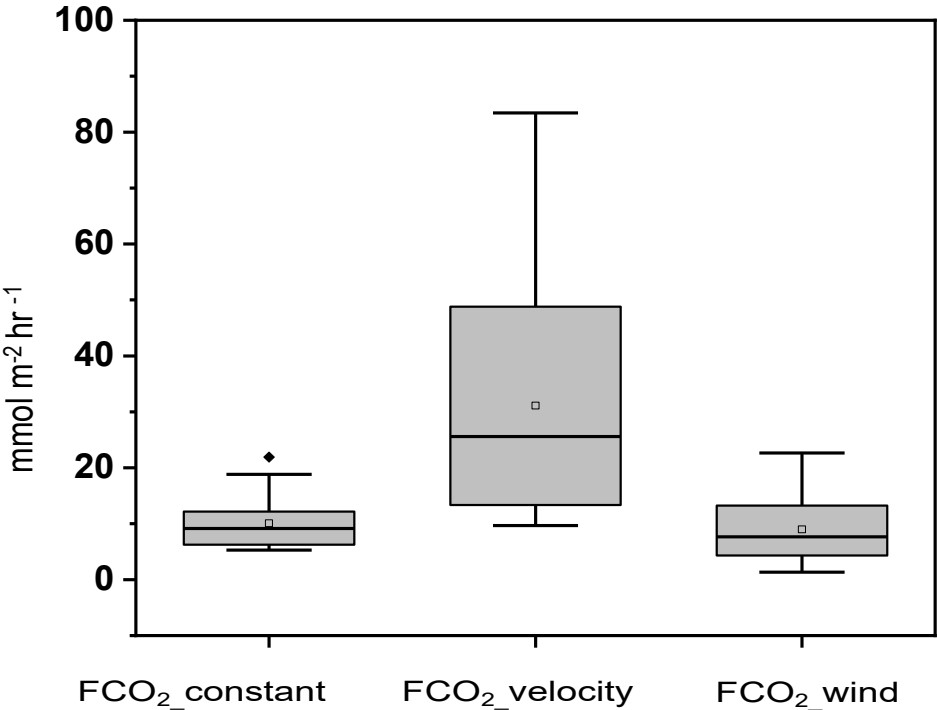

**Figure 5.** Large variation in velocity-based FCO$_2$ (mmol m$^{-2}$ h$^{-1}$) when compared to those with constant FCO$_2$ and wind-based FCO$_2$ during the period of 2019–2021. The points inside boxplots indicate the mean values for the parameters.

The $CO_2$ flux for fixed and wind-dependent $K_T$ showed a similar kind of rate and pattern, whereas velocity-dependent $K_T$ showed a higher $CO_2$ flux rate in the lower Mississippi River. A positive correlation was found between $K_T$ and $CO_2$ flux, where the pattern is more noticeable in $CO_2$ outgassing for velocity-based $K_T$ (Figure 6).

$CO_2$ outgassing estimates were largely different among the three $K_T$ methods, as an ANOVA test demonstrated a $p$ value less than the significance level, 0.05. There was a significant variation in FCO$_2$ due to velocity-based $K_T$, which is linked to discharge fluctuations. This significance was checked using a multiple-comparison by Tukey HSD test, as shown in Table 4. Lower $p$ values (<0.05) found in flux differences between FCO$_2$ (velocity $K_T$) and FCO$_2$ (constant $K_T$) and between FCO$_2$ (wind $K_T$) and FCO$_2$ (velocity $K_T$) showed significance (Table 4). On the other hand, the flux difference between FCO$_2$ (wind $K_T$) and FCO$_2$ (fixed $K_T$) was found to be insignificant ($p > 0.05$) (Table 4).

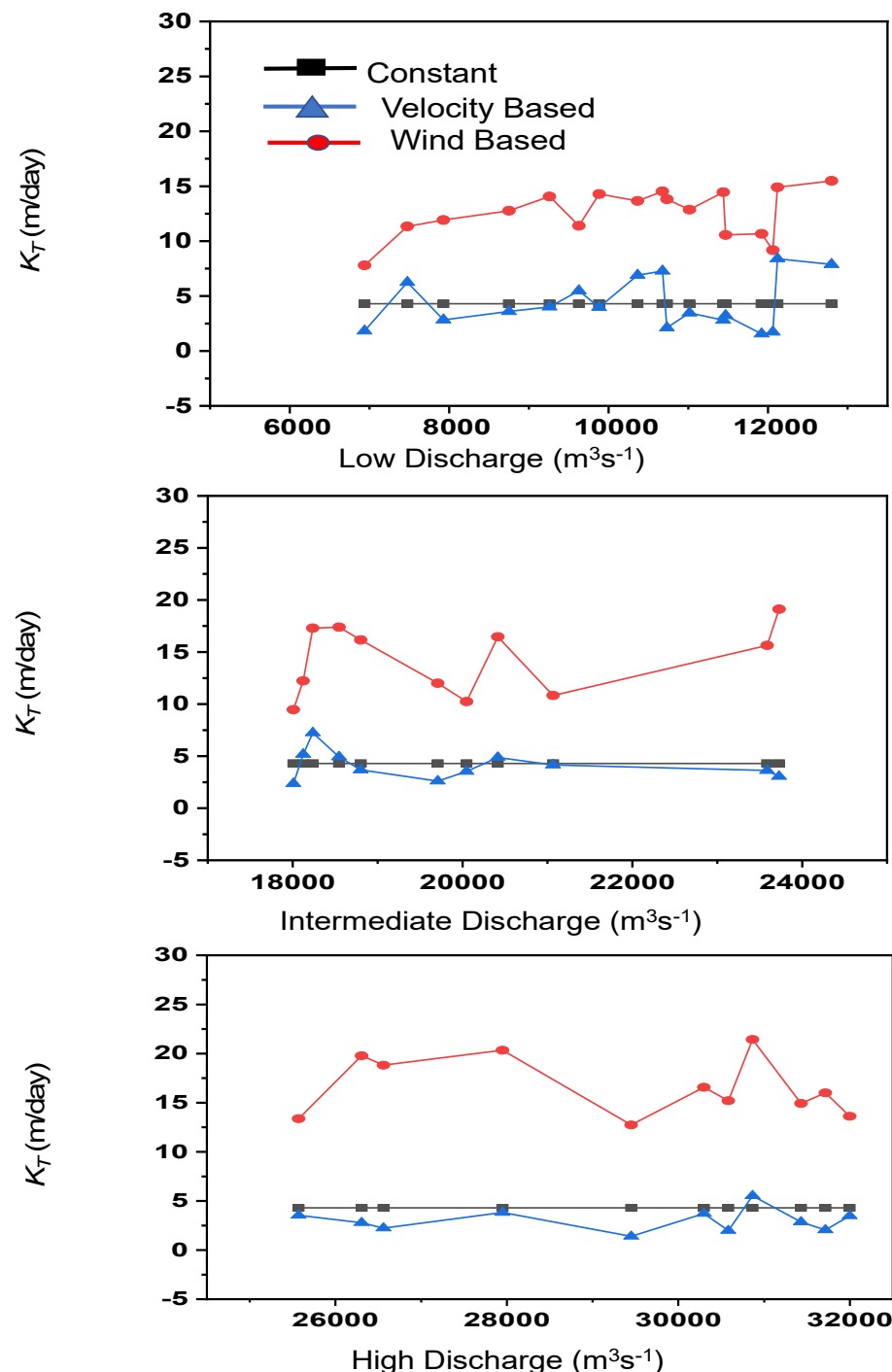

**Figure 6.** Changes of three different gas transfer $K_T$ values with discharge of the Mississippi River at Baton Rouge.

**Table 4.** A multiple-comparison by Tukey HSD test (conf. level = 0.95) pairwise between different $CO_2$ fluxes for different types of $K_T$ values showed higher significant variance in the velocity-based method than the wind-based method.

|  | Differences | Lower | Upper | *p* |
|---|---|---|---|---|
| $FCO_2$\_velocity-$FCO_2$\_constant | 32.71 | 23.15 | 42.27 | *p* < 0.05 |
| $FCO_2$\_wind-$FCO_2$\_constant | −1.59 | −11.15 | 7.97 | *p* > 0.05 |
| $FCO_2$\_wind-$FCO_2$\_velocity | −34.30 | −43.86 | −24.74 | *p* < 0.05 |

## 4. Discussion

### 4.1. Variability in Velocity-Based Estimation of $K_T$

This study demonstrates that the estimation of $CO_2$ outgassing at the water–air interface is strongly affected by how the gas exchange coefficient is determined. Even though this has been reported by previous studies, the magnitude of the influence is much greater than expected. The greatest variable $K_T$ found in this comparative study is the velocity-based estimation. In the case of velocity-based $K_T$, the observed wide variation can be partially attributed to fluctuations in river discharge. In the lower Mississippi River, higher $K_T$ values were observed with increasing discharge values (Figure 6). The Mississippi River exhibits a dynamic flow regime subject to fluctuations in discharge, varying from 6737 to 36,811 $m^3 s^{-1}$ (Figure 2), which are influenced by a range of factors, including precipitation, snowmelt, and anthropogenic interventions such as dam operations. The study period was marked by an unusual flood event in 2019, characterized by high flows. Additionally, 2021 was recorded as one of the warmest years in the Mississippi River Basin and exhibited above-average annual precipitation [29]. The occurrence of severe weather extremes throughout the year has led to conditions that surpass the long-term average. These higher discharge levels may yield increasing flow velocities, whereas reduced discharge levels may give rise to lowered velocities.

Several studies have highlighted the impact of discharge variability on the carbon dioxide dynamics within river networks. Ran et al. identified the large carbon fluxes in high-velocity zones with substantial soil erosion or extensive rock weathering, which mobilize organic carbon into the river network [40]. Reiman and Xu also found a positive linear association between river discharge and $CO_2$ outgassing in the Mississippi River, which is consistent with our findings [15]. Along with the discharge, the flow velocity can vary due to the spatial and physical variations in the structure along the lateral and latitudinal dimensions of a river [41]. The Mississippi River displays variations in channel dimensions, depths, and geomorphic characteristics across its entire course. A shallow stream section increases surface turbulence that improves gas exchange at the water–air interface, which is consistent with our study's findings on variations in discharge velocity. Levee construction may be another factor contributing to high velocity and greater $K_T$ and carbon fluxes.

### 4.2. Marginal Effect of Wind Speed on $K_T$ Estimation

We found that $CO_2$ outgassing is linearly correlated with $K_T$ when its values are below approximately 10 m/day (Figure 7). Above 10 m/day, the relation between $K_T$ and $FCO_2$ becomes exponential. The flow velocity contributes approximately three times more $K_T$ than the constant and wind (Table 2). This explains why velocity-based $K_T$ produces much greater $CO_2$ outgassing.

The carbon flux is in linear relation with $K_T$ below 10 m/day (Equation (10)), while an exponential relationship can best present the relation for $K_T$ values above 10 m/day with Equation (11):

$$Y = -0.143 + 2.850\,x,\ R^2 = 0.280 \tag{10}$$

$$Y = 2.597\,e^{0.178\,x},\ R^2 = 0.731 \tag{11}$$

where $FCO_2$ is the dependent variable, and $K_T$ is the independent variable. $R^2$ is the coefficient of determination, which indicates the proportion of the variance in the dependent variable that can be explained by the independent variables. The $R^2$ value indicates how well the model fits the data. After comparing it to a linear function, we have chosen the relationship as an exponential function. The exponential equation had a higher $R^2$ value (0.73) than the linear equation (0.48).

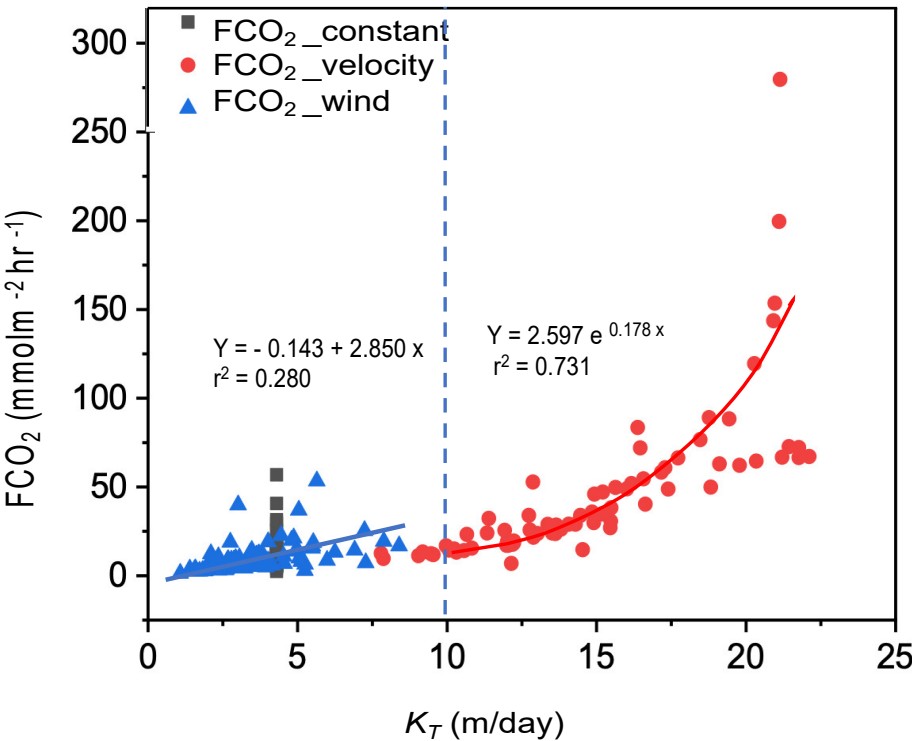

**Figure 7.** Relationship of estimated $CO_2$ fluxes ($FCO_2$) (mmol m$^{-2}$ h$^{-1}$) with different $K_T$ (m/day) calculated from velocity-based and wind-speed-based methods, as well as a constant. The vertical dashed line indicates a threshold of $K_T$ at approximately 10 m/day, above which $FCO_2$ increases exponentially.

The marginal effect of wind speed on $K_T$ found in this study results in the lower carbon flux estimates in the Mississippi River for the wind-based estimation. Lower wind speeds along the Mississippi River may restrict the extent of turbulent mixing, resulting in less gas exchange and, consequently, lower $K_T$ values. The wind has been recognized as a significant contributor to surface turbulence, particularly in lakes and estuaries where wind speed is the primary turbulence producer [31,40,42]. In contrast to the lentic system, i.e., Lake System, where wind-induced turbulence can have a greater impact on gas exchange, large rivers tend to be less responsive to wind-driven processes. Their carbon dynamics are governed predominantly by river discharge, temperature, and the transport of organic matter from floodplains. Previous research has also demonstrated the marginal effect of wind speed on carbon fluxes in large streams [31,40,42]. A previous study has found that wind stress dominates turbulence in the surface aqueous boundary layer for systems with depths greater than 10 m or at wind speeds greater than 8 ms$^{-1}$ [32]. Our findings with wind range (0.4 to 4 ms$^{-1}$), which is much lower than lakes and estuaries, are found to be consistent with these prior observations, suggesting that wind speed in large rivers such as the Mississippi may have a limited influence on carbon fluxes relative to other driving factors.

### 4.3. $K_T$ and $FCO_2$ Differences with Previous Studies

Previous studies on $CO_2$ outgassing in the Mississippi River used a constant $K_T$ value. For instance, the study by Xu and Xu [28] used 4.3 m/day, while the studies by Potter and Xu [29], Reiman and Xu [15], and Dubois et al. [20] used 3.9 m/day (Table 5). The $FCO_2$ fluxes reported from these studies are generally lower than those found in our study. The carbon flux resulting from velocity-functioned $K_T$ yields an average of 4667 g C m$^{-2}$ yr$^{-1}$, which is five times greater than those by Potter and Xu [29] with their estimation of 864.6 g C m$^{-2}$ yr$^{-1}$, six times greater than those of Xu and Xu [28] with their estimation of 777 g C m$^{-2}$ yr$^{-1}$, seven times greater than those of Reiman and Xu [15] with their estimation of 654 g C m$^{-2}$ yr$^{-1}$, and 4.5 times greater than those of Dubois et al. [20] with

their estimation of 1036 g C m$^{-2}$ yr$^{-1}$ (Table 5). The large discrepancies in carbon flux estimations highlight the critical importance of selecting a proper gas transfer coefficient.

**Table 5.** Comparison of gas transfer velocity ($K_T$) and $CO_2$ outgassing (FCO$_2$) estimated from this study from previous studies in the lower Mississippi River. All numbers from the 2019 flood research are sums or averages from the 212 days of flooding. The values cited from other research or this study are either the annual (365-day) totals or the averages for the respective time.

| Study Period | Q | *p*CO$_2$ | FCO$_2$ | $K_T$ | References |
|---|---|---|---|---|---|
| | km$^3$ yr$^{-1}$ | µatm | g C m$^2$ yr$^{-1}$ | m/day | |
| 2019–2021 | 637 | 2139 ± 1454 | 1224 | **4.3** | |
| | | | 4667 | $K_T$_velo 14.6 ± 3.8 | This Study |
| | | | 1057 | $K_T$_wind 3.65 ± 1.60 | |
| 2021 | 500 | 1703 ± 646 | 864.6 | 3.9 | [29] |
| 2019 Flood | 634 | 2217 ± 805 | 777 | 4.3 | [28] |
| | | | 705 | 3.9 | |
| 2015–2018 | 548 | 1500 ± 743 | 654 | 3.9 | [15] |
| 2000–2001 | 374 | 1363 ± 267 | 1036 | 3.9 | [20] |

Studies on $K_T$ effects from other rivers also showed a large variation depending on the estimation method [33,40]. Therefore, to accurately assess carbon flux in riverine systems, it is essential to consider the gas exchange coefficient and their spatiotemporal variability.

*4.4. Limitations and Future Implications*

This study is a numeric assessment of the effect of three calculation methods for $K_T$ on $CO_2$ outgassing estimation. The Mississippi River is used here only as a case study because of the availability of its longer-term field measurements on *p*CO$_2$, wind speed, temperature, and other environmental parameters. The findings obtained from this methodological study suggest that future research on riverine $CO_2$ outgassing using $K_T$ estimation should be cautious in the context of large variation and uncertainty. Therefore, the findings are not geographically limited and can be applicable to all future studies using these methods to obtain $K_T$.

This assessment on the impact of flow velocity and wind speed on gas exchange is solely numeric, i.e., not using field-based measurements of gas transfer coefficient $K_T$. Therefore, the assessment reflects the parameter calculation applying a numerical calculation method rather than the direct, field-measured $K_T$. Theoretically, $CO_2$ outgassing (FCO$_2$) estimates can be improved by using field-measured $K_T$ for direct flux-measuring techniques, such as the floating chamber approach. If this were to occur, it would result in a more transparent comprehension of the connection between discharge velocity and $CO_2$ outflow. However, using inaccurate $K_T$ values may result in considerable bias in carbon flux estimations, impeding the accurate evaluation of carbon budgets and hindering our comprehension of the carbon cycle in aquatic ecosystems. This may affect our capacity to keep track of and regulate carbon dynamics, evaluate the effects of human activity on carbon emissions, and make sound decisions regarding strategies for coping with climate change. However, the reality is that field measurements on $K_T$ are not only expensive and time-consuming but also may not be representative of large rivers in different channel form reaches and under different flow conditions. Although the actual variation of $K_T$, along with flow velocity (i.e., discharge) and wind speed, can only be verified through field measurement, this study does provide crucial insights into the two factors' influence on $CO_2$ emission estimation with the current approaches. The selection of appropriate $K_T$ for measuring carbon flux has significant future implications for improving measurement techniques and advancing more precise models. Integrating advanced measurement techniques, such as eddy covariance and stable isotope analysis, along with the incorporation

of high-resolution spatial and temporal data, can significantly improve the accuracy of estimating $K_T$ and carbon flux.

## 5. Conclusions

This study assessed the impact of river flow and wind speed on a key parameter used in riverine $CO_2$ outgassing prediction, the gas exchange coefficient $K_T$. As a case study, we utilized 3-year (2019–2021) field measurements on the partial pressure of dissolved carbon dioxide in the lower Mississippi River, near its mouth to the Gulf of Mexico. Over the study period, the river discharge fluctuated greatly from 6938 to 36,812 $m^3 s^{-1}$, resulting in a velocity range from 0.8 to 1.9 $ms^{-1}$. The large variation in flow condition strongly affected the computational results of $K_T$, which ranged from 7.80 to 22.11 m/day. Wind speed also varied greatly, i.e., from 0.4 to 4 $ms^{-1}$, but its effect on $K_T$ was marginal, ranging from 0.77 to 8.40 m/day. The largest variation of carbon outgassing rates was found in the velocity-based $K_T$ method, namely from 6.8 to 280 mmol $m^{-2} h^{-1}$. Consequently, the most variable carbon flux rate (FCO$_2$) was found in the velocity-based $K_T$ method (range: 6.8 to 280 mmol $m^{-2} h^{-1}$, mean: 44.36 mmol $m^{-2} h^{-1}$) when compared with those in the wind-based method (range: 1.32–53.40 mmol $m^{-2} h^{-1}$, mean: 10 mmol $m^{-2} h^{-1}$) and the constant $K_T$ of 4.3 (range: 2.42–56.87 mmol $m^{-2} h^{-1}$, mean: 11.64 mmol $m^{-2} h^{-1}$). Considering that river channel geometry changes spatially across the landscape, $CO_2$ outgassing from the river can, therefore, vary substantially. Based on these findings, we conclude that the effect of river channel geometry and flow velocity on $CO_2$ outgassing is still largely underestimated, resulting in substantial uncertainty in the estimation of carbon flux.

**Author Contributions:** Writing—original draft preparation, A.D. and Y.J.X.; writing—review and editing, Y.J.X.; data curation, A.D. and Y.J.X.; conceptualization, Y.J.X.; funding acquisition, Y.J.X. All authors have read and agreed to the published version of the manuscript.

**Funding:** This research was supported by the United States Geological Survey through the Water Resources Research Act Program Annual Base Grants (G21AS00517) and a US Department of Agriculture Hatch Fund project (Project#: LAB94459).

**Data Availability Statement:** River discharge and stage data used in this study can be obtained from the United States Geological Survey (https://waterdata.usgs.gov/nwis/sw, accessed on 18 July 2023). Field measurement data during this study and the analysis data are available from the corresponding author upon reasonable request.

**Acknowledgments:** The authors sincerely thank Jeremy Reiman and Zhen Xu for their outstanding assistance in field measurements on river $pCO_2$. The authors appreciate the U.S. Geological Survey for making the river discharge and stage data available for this study. The authors are also thankful to the associate editor and three anonymous reviewers for their valuable comments and suggestions, which have been helpful for improving the quality of this manuscript.

**Conflicts of Interest:** The authors declare no conflict of interest.

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
