# Peer review of "Large Uncertainties in CO2 Water–Air Outgassing Estimation with Gas Exchange Coefficient KT for a Large Lowland River"

_water, doi:10.3390/w15142621_

Round 1

Reviewer 1 Report

Large uncertainties in CO2 water-air outgassing estimation with gas exchange coefficient KT for a large lowland River

The subject of this manuscript is very interesting for readers. The novelty of this study is acceptable. It needs to following minor modifications:

1- I advise that authors add new references (from 2021 to 2023) to manuscript.

2- Authors should state hydrological parameters in the Baton Rouge in Louisiana such as elevation, the mean annual precipitation, evaporation and temperature.

3- Authors did not state the mean annual wind speed and the partial pressure of dissolved carbon dioxide.

4- Figure 1 has not longitude and latitude.

The quality of English language is good.

Reviewer 2 Report

1.       The gas exchange coefficient (KT) should be defined clearly and its importance explained at the beginning of the study. It is essential to provide a comprehensive understanding of KT and its relevance to the research. Additionally, the use of a fixed value of 4.3 m/d in the study should be justified and discussed in terms of its appropriateness for the research objectives and the specific context of the study. Subsection 4.3, which concludes the need for selecting the proper KT, should be moved to the introduction section to establish its significance and relevance early on in the article.

2.       In Figures 2 and 3, it is recommended to add more points to the x-axis to provide a clearer representation of the data and improve the visual interpretation. What are the points inside the boxplots (Figure 4)?

3.       In line 232, the authors mentioned the exclusion of extreme outliers. It would be beneficial to refer to the figures showing the outliers and provide detailed information about these outlier points. This includes explaining the reasons for their exclusion and their potential impact on the analysis. Clarifying whether these outliers are important or if they occurred due to any mistakes is necessary.

4.       Throughout the text, all equations, figures, and tables should be explicitly mentioned and referred to using consistent formatting and style. For instance, in line 198, the authors should mention the equation number instead of using the phrase "equation below." This consistency should be maintained throughout the article.

5.       In line 358, the authors should explain how the significant variation was determined, providing details about the statistical methods, tests, or analyses used to assess the significance.

6.       In Table 4, the figures should be rounded to a maximum of three decimal places, ensuring consistency throughout the table. The meaning of "P" in this table should be clarified, specifying that it represents the p-value. Instead of displaying P=0.0000, it should be stated as "P<0.05" or similar, indicating that the p-value is less than the considered significance level (e.g., 0.05).

7.       In line 390, where the authors conclude an exponential relationship, it is important to clarify whether they compared it with a linear relationship or other types of relationships. The meaning of "r2" should also be explained, specifying that it represents the coefficient of determination. Additionally, the authors should utilize functions outside of the figure with proper descriptions to enhance clarity and facilitate interpretation.

8.       The article exhibits inconsistencies in citation style. In lines 106, 104, and 420, different citation styles are observed, including the use of "others" instead of "et al." It is important to ensure consistency in the citation style throughout the manuscript.

9.       Several typographical mistakes have been identified. Line 30 shows a change in font size.

10.   Line 195, KT and K600. Line 198 is starting with ‘and’. Line 305 ‘Eqs’. Line 347, ‘A multiple comparisons’? Line 400, ‘functioned estimation’? Line 445, ‘algorisms’? What is the meaning of this line? Authors have not explained the ‘algorithm’ clearly earlier. 

Reviewer 3 Report

General Comments: The study is focused on numeric assessment of the impact of river flow velocity and wind speed on KT and the consequent CO2 emission rate. No doubt, the study is aligned with the scope of the journal and provides some interesting insights into the literature.

However, for scientific contribution to literature, the study needs quality improvement. Below are my comments and suggestions that the authors need to incorporate.

Main Points:

1.     I found a weakness in the identification of the research gap identification paragraph. There are many studies cited in the papers have already been published in the literature. How the author distinguished this research has significance.

2.     Authors need to specify the location of the experimental area of the study and answer the question do the results of this study can be integrated into other regions of the world that have different climate conditions.

3.     Does the author don’t think that there would be an environmental change impact as well..?

4.     Results are well written; however, I suggest using reader-friendly language as the reader can easily understand and get more benefit from the research output.

5.     Conclusion section is too large and needed to be precise.

Minor Points:

I found a few grammatical mistakes in the manuscript

Overall English is well written. However, I found a few grammatical mistakes in the manuscript

Round 2

Reviewer 2 Report

The authors have incorporated the suggestions and now the manuscript has improved significantly as compared to earlier version.

Reviewer 3 Report

The author has put a lot of energy into the revised manuscript. The revised manuscript is improved. The quality of the revised manuscript is now seeming satisfactory for publication in the journal. However, I strongly recommend the author to cite research article, and scientific research report to be technically sound.

l  The author cited more reports I would suggest citing the following recent studies on dealing with similar topics and streamflow in the river basins.

i.                https://doi.org/10.3390/w11010043

ii.               https://doi.org/10.1007/s12205-022-0650-z

iii.              https://doi.org/10.1071/MF21154

iv.             https://doi.org/10.1007/s12205-021-0151-5